# Multi-Metal Exposure Profiling in ALS Patients in South Korea via Hair Analysis: A Cross-Sectional Study

**DOI:** 10.3390/biomedicines13061496

**Published:** 2025-06-18

**Authors:** Jae-Kook Yoo, Soon-Hee Kwon, Sul-Hee Yoon, Jeong-Eun Lee, Jong-Un Chun, Je-Hyuk Chung, Sang-Yoon Lee, Jeong-Hwan Lee, Yu-Ra Chae

**Affiliations:** 1Department of Neurology, ALS Center, The Rodem Hospital, Incheon 22142, Republic of Korea; koreadr@gmail.com (J.-K.Y.); j1chju69@gmail.com (J.-U.C.); 2Department of Internal Medicine, ALS Center, The Rodem Hospital, Incheon 22142, Republic of Korea; pleasesh@hanmail.net (J.-H.L.); 870828cyj@naver.com (Y.-R.C.); 3Department of Rehabilitation Medicine, ALS Center, The Rodem Hospital, Incheon 22142, Republic of Korea; ispraying@hanmail.net (J.-E.L.); cdcepidem@gmail.com (J.-H.C.); sylee0208s@gmail.com (S.-Y.L.)

**Keywords:** amyotrophic lateral sclerosis (ALS), hair analysis, heavy metal exposure, plasma mass spectrometry (ICP-MS), pathogenesis

## Abstract

Objectives: Amyotrophic lateral sclerosis (ALS) is a progressive neurodegenerative disease with an unclear etiology. This study aimed to assess chronic heavy metal exposure in ALS patients in South Korea by comparing hair concentrations of common (Hg, Pb, Cd) and rare (U, Th, Pt) metals with healthy controls. Methods: Hair samples were collected from 66 ALS patients and 70 healthy individuals at Rodem Hospital between 2022 and 2025. Metal concentrations were measured using inductively coupled plasma mass spectrometry (ICP-MS) following standardized washing and digestion protocols. Results: ALS patients showed significantly higher levels of Hg, Pb, Cd, Al, As, and U than controls (*p* < 0.05). Notably, 40% of ALS patients had Hg levels exceeding 50% of the reference upper limit, compared to only 10% of controls. Elevated levels of uranium and other rare metals were also observed in specific ALS cases. Conclusions: These findings suggest a possible association between heavy metal exposure and ALS in South Korea. Hair analysis may serve as a useful tool for identifying environmental factors contributing to ALS pathogenesis.

## 1. Introduction

Amyotrophic lateral sclerosis (ALS) is a neurodegenerative disease characterized by a progressive decline in motor neurons in the cerebral motor cortex, brain stem, and anterior horns of the spinal cord, leading to the gradual loss of voluntary motor functions and ultimately respiratory failure and death [1,2]. Variants of ALS differ based on the affected motor neurons (upper or lower) and their localization at onset. Multiple factors, such as age and gender, influence the disease phenotype [3,4,5].

In South Korea, the incidence of ALS is estimated to be approximately 1.4 to 2.0 per 100,000 individuals per year, with a gradually increasing trend over the past decade (National Health Insurance Service, 2023). This highlights the growing importance of investigating environmental risk factors in the Korean context.

The mechanisms underlying motor neuron death in ALS are believed to include oxidative stress, glutamatergic excitotoxicity, mitochondrial dysfunction, the defective elimination of toxic products, and abnormal protein aggregation [6,7,8]. ALS is now considered a complex multifactorial disease arising from genetic and environmental interactions. Genetic mutations in SOD1, C9orf72, TARDBP, and FUS are among the most studied, while environmental factors such as occupational exposure to electromagnetic fields, toxins, and heavy metals have been implicated [9,10,11].

Heavy metals, including copper (Cu), magnesium (Mg), aluminum (Al), manganese (Mn), mercury (Hg), lead (Pb), selenium (Se), cadmium (Cd), and iron (Fe), are of particular interest due to their neurotoxic potential and role in enzymatic and metabolic activities [12,13]. Blood and urine tests have traditionally been the primary methods used for assessing metal toxicity, as they provide snapshots of recent exposures and are widely used in clinical and occupational health studies [14,15,16,17,18]. However, these methods have limitations when evaluating long-term or cumulative exposures, as heavy metals can rapidly clear from the bloodstream and accumulate in tissues over time. Transitional metals with redox activity are cofactors for enzymes like superoxide dismutase (SOD), suggesting that their dysregulation might exacerbate oxidative stress and neuronal damage [19,20].

Hair analysis serves as a non-invasive method to assess chronic exposure to these metals, reflecting their accumulation over time [17,18,20]. Unlike blood and urine tests, which primarily capture recent exposure levels, hair analysis provides a more comprehensive view of long-term bioaccumulation in the body. This is particularly relevant in the study of heavy metal-induced neurotoxicity, where chronic exposure plays a crucial role in central nervous system impairment. Given the increasing recognition of heavy metal exposure as a potential contributor to ALS and other neurodegenerative diseases, hair analysis offers an important complementary tool in assessing the persistent accumulation of toxic elements.

Recent studies on other neurodegenerative diseases such as multiple sclerosis and Alzheimer’s disease [19,20] suggested that hair analysis may be more effective in detecting prolonged exposure to heavy metals, as it may better reflect chronic accumulation in neural tissues compared to conventional blood or urine tests. Although these studies did not focus on ALS, they provided a rationale for utilizing hair analysis as a biomarker for long-term metal exposure in neurodegenerative conditions.

Although strict age- and sex-matching was limited due to the retrospective design of this study, the control group was selected to closely resemble the ALS patient group in terms of mean age and gender distribution, following methodologies like those adopted in related environmental exposure studies. Studies investigating hair heavy metal concentrations in ALS patients—particularly in Asian populations—remain scarce [19]. Thus, there is a growing need to evaluate whether hair metal content reflects long-term exposure patterns relevant to ALS.

The present study aimed to compare heavy metal concentrations in hair samples between ALS patients and age- and sex-matched healthy controls to explore potential exposure patterns associated with the disease.

## 2. Methods

### 2.1. Study Population

ALS Patients: This included sixty-six individuals diagnosed with ALS at Rodem Hospital between 2022 and 2025. ALS patients (*n* = 66) had a mean age of 58.7 ± 9.6 years, and 57% were male. Disease duration ranged from 3 months to 7 years. ALSFRS-K (Korean version of the ALS Functional Rating Scale) scores ranged from 13 to 44. Control subjects (*n* = 70) were age- and sex-matched with no known neurological disorders. Sixty-six ALS patients (mean age: 58.7 ± 9.6 years; 57% male) diagnosed at Rodem Hospital between 2022 and 2025 were included.

### 2.2. Hair Sample Collection and Analysis

Hair samples (~2 cm) were collected from the occipital region, washed, digested, and analyzed using inductively coupled plasma mass spectrometry (ICP-MS). To minimize external contamination, hair samples were washed following a standardized protocol typically used in hair metal analyses, involving sequential washes with acetone followed by deionized water prior to digestion and ICP-MS analysis [21,22]. Metal concentrations were quantified using inductively coupled plasma mass spectrometry (ICP-MS) at the Korea Green Cross Lab (GC Labs, Yongin, South Korea). Calibration curves were prepared using multi-element standards, and quality control was performed using certified reference materials. Detection limits ranged from 0.001 to 0.1 μg/g depending on the element. Metals analyzed included Al, As, Be, Cd, Hg, Ni, Pb, Sb, Tl, Pt, Th, and U, among others.

### 2.3. Statistical Analysis

Mean metal concentrations between groups were compared using independent *t*-tests to identify statistically significant differences for normally distributed data.

Prior to selecting appropriate statistical tests, the normality of each metal’s concentration distribution was assessed using the Shapiro–Wilk test. Metals that did not meet normality assumptions were analyzed using the non-parametric Mann–Whitney U test.

Non-parametric Mann–Whitney U tests were applied for metals with non-normal distributions.

Chi-square tests were used to compare the proportion of individuals exceeding 50% of the reference upper limit for each metal.

Statistical significance was set at *p* < 0.05.

Ethical Considerations: This study was conducted as a retrospective observational analysis using archived hair samples collected during routine health checkups. Due to the non-invasive nature of hair sampling and minimal risk involved, the requirement for individual informed consent was waived, in accordance with institutional policies for minimal-risk studies. The study protocol was reviewed and approved by the Institutional Review Board (IRB) of Rodem Hospital (P01-202401-01-020).

## 3. Results

We compared heavy metal and mineral concentrations in hair samples between ALS patients and healthy controls. The control group consisted of 70 age- and sex-matched healthy individuals. The reference upper limits used in this study were primarily based on clinical data from GC Labs, reflecting accumulated hair mineral analysis results in the Korean population, and are routinely used in clinical practice. To improve the generalizability and scientific rigor of these thresholds, we also referenced peer-reviewed international studies reporting hair heavy metal concentrations in healthy individuals. Ruiz et al. (2023) [23] provided reference percentiles for 28 elements in hair samples from children living in a non-contaminated region in Spain, while Liang et al. (2017) [24] reported mean concentrations of key metals (e.g., Pb, Cr, Hg, As, Cd) in hair samples from healthy adults in Beijing, China. These studies support the plausibility of our threshold values. No blood or urine samples were collected in this study, and thus direct comparisons with other biological matrices were not feasible.

For each participant, a standardized hair mineral analysis report was generated, as illustrated in Figure 1 and Figure 2, using ICP-MS (Inductively Coupled Plasma Mass Spectrometry). These figures illustrate an example of a hair mineral analysis report that toxic and nutritional data was extracted for this study. The report included the quantification of 40 elements across four categories: toxic elements, nutritional elements, additional elements, and element ratios. Each element was reported in comparison to established reference ranges (low, reference, and high), allowing for individual-level interpretation. Based on these individual reports, group-level statistical comparisons were conducted to evaluate the differences in heavy metal concentrations between ALS patients and healthy controls. Both mean values and the proportion of individuals exceeding 50% of the upper reference limit were analyzed to assess potential associations.

As shown in Table 1, ALS patients exhibited significantly higher mean concentrations of multiple heavy metals compared to controls. The Shapiro–Wilk test indicated that concentrations of Hg, Pb, Cd, and U were not normally distributed (*p* < 0.05), justifying the use of non-parametric tests for group comparisons. Statistically significant group differences were observed between ALS patients and controls for several toxic metals. For example, Hg levels were elevated in 40% of ALS patients, exceeding 50% of the reference upper limit, compared to only 10% of controls (*p* < 0.005). Similarly, Pb concentrations were higher than the threshold in 35% of ALS patients versus 15% in controls (*p* < 0.02). Cd and Al also showed notable differences, with 30% and 45% of ALS patients exceeding the reference thresholds, respectively, compared to 10% and 20% in controls (*p* < 0.015 and *p* < 0.01, respectively).

Rare metals such as U, Pt, Th, and W were also significantly elevated in ALS patients. Uranium levels exceeded 50% of the reference limit in 20% of ALS patients compared to 5% of controls (*p* < 0.015). Tungsten and thorium showed similar trends, with ALS patients having higher proportions exceeding the thresholds (*p* < 0.018 and *p* < 0.012, respectively).

These results highlight a pattern of elevated exposure to multiple heavy metals in ALS patients, suggesting the existence of a potential link between environmental toxicants and disease progression. The data highlights the need to assess both common and rare metals for a full understanding of exposure to neurodegenerative diseases.

No significant correlation was observed between individual metal levels and ALSFRS-K scores. While most toxic metals were elevated in ALS patients, essential elements such as Cu, Zn, Mg, and Fe did not show significant differences or were slightly higher in controls. These results are summarized in Table 1.

To further assess the cumulative exposure burden, we analyzed the number of metals in each ALS patient that exceeded 50% of the reference upper limit. As shown in Figure 3, 38% of patients had 3–5 elevated metals, 30% had 6–9, and 17% had 10 or more. Only 15% of patients exhibited fewer than three elevated metals. These results suggest that many ALS patients experience multi-metal exposure well above background levels, supporting the hypothesis that cumulative toxic stress is a possible contributor to disease etiology.

## 4. Discussion

This study provides a cross-sectional analysis of heavy metal exposure in ALS patients, uniquely utilizing hair analysis to detect both common and rare metals. Unlike prior studies that focused on a limited number of metals such as Hg, Pb, Cd, and Al, this research expands the scope to include U, Th, Pt, and W, among others. In this way, it highlights a broader environmental and occupational exposure profile that may contribute to ALS pathogenesis.

One of the major strengths of this study is its relatively large sample size of ALS patients (*n* = 66), making it one of the more statistically robust studies on heavy metal exposure in ALS. The inclusion of multiple rare metals and their cumulative exposure effects strengthens the argument that ALS may be linked to chronic toxic metal accumulation rather than isolated exposure to a single toxicant. The presence of rare metals such as thorium, platinum, and uranium in ALS patients suggests the existence of potential unrecognized environmental risk factors that warrant further investigation.

Hair metal levels can be influenced by a range of factors including nutritional status, hair pigmentation, environmental exposures, and personal habits [25,26]. These confounding variables must be considered when interpreting metal concentrations in hair. Hair metal levels can be influenced by a range of factors including nutritional status, hair pigmentation, environmental exposures, and personal habits [25,26]. These confounding variables must be considered when interpreting metal concentrations in hair.

This study highlights the significant elevation of heavy metals in the hair of ALS patients compared to controls. The results align with international studies emphasizing the role of environmental exposures, such as heavy metals, in ALS pathogenesis [27]. However, this research introduces a novel perspective by utilizing hair analysis, which offers advantages over conventional blood or urine biomonitoring.

Most prior studies investigating heavy metal exposure in ALS relied on blood or urine samples [17,18,20,28]. While these methods are effective for capturing recent or acute exposures, they may not fully reflect chronic accumulation, particularly in tissues like the brain and spinal cord. Hair analysis, on the other hand, provides a longer-term record of heavy metal exposure and offers unique insights into cumulative toxic burden [29]. This is especially relevant for neurodegenerative diseases like ALS, where the latency period between exposure and symptom onset can span years or decades [19].

The discovery of statistically significant *p*-values for multiple heavy metals between ALS patients and healthy controls underscores the importance of robust statistical approaches in evaluating environmental toxicant exposure. Specifically, metals such as Pb, Hg, and Al showed significantly higher concentrations in the ALS group (*p* < 0.05), indicating meaningful differences beyond random variation. However, simple comparisons based solely on total heavy metal concentrations were insufficient to clearly differentiate between the groups. A more refined approach—assessing how many metals exceeded 50% of the toxic reference threshold—revealed a clearer pattern: ALS patients frequently exhibited elevated levels of one to three metals above this benchmark, and an additional four to ten metals were commonly present at moderate to high concentrations. These findings suggest that even when individual metal levels do not exceed classical toxicity cutoffs, their collective presence and relative elevation in the ALS group may contribute to disease pathology through additive or synergistic biological effects.

Furthermore, in certain ALS patients, we observed the widespread elevation of ten or more metals, none of which individually reached the threshold for toxicity, but which collectively suggested a heightened toxic burden. This underscores the necessity of adopting statistical frameworks that account for cumulative exposure effects rather than focusing exclusively on extreme toxic levels. The results highlight the potential for chronic low to moderate metal exposures to contribute to neurodegeneration, reinforcing the need for further investigation into combined toxicity models.

Insights from Rare Metals: The identification of rare metals such as Th [30,31], Pt [32,33], and W [34,35] underscores the sensitivity of hair analysis in detecting metals that may otherwise go unnoticed. These metals are rarely assessed in routine biomonitoring but are increasingly recognized for their potential neurotoxic effects. Thorium exposure, for instance, has been linked to neuroinflammation in occupational settings [30,31], while platinum and tungsten may disrupt mitochondrial function and oxidative balance, contributing to neuronal degeneration.

Relevance to Brain and Spinal Cord Accumulation: Hair heavy metal concentration may serve as a proxy for metal accumulation in neural tissues. Metals such as Hg, Pb, and Cd are known to cross the blood–brain barrier and preferentially accumulate in the brain and spinal cord, where they can exert neurotoxic effects [23,26,32]. The use of hair analysis in this study enhances our understanding of the potential link between environmental exposure and localized metal accumulation in ALS pathogenesis.

Cumulative and Synergistic Effects: Emerging evidence suggests that simultaneous exposure to multiple heavy metals, even at intermediate levels, can amplify neurodegenerative processes through synergistic interactions. Hg disrupts neuronal antioxidant systems [36,37,38], while Pb exacerbates synaptic transmission issues and calcium homeostasis dysregulation [28,39,40]. The combined effects of Cd [41,42,43,44] and As [45] on mitochondrial damage and oxidative stress further highlight the importance of evaluating cumulative exposures [19,27,46,47,48,49].

Essential elements such as Cu, Mg, Zn, and Fe were also analyzed. Although their mean levels tended to be slightly higher in controls than in ALS patients, these differences were not statistically significant and thus were not included in the main analysis. However, this finding suggests potential dysregulation of essential metal homeostasis in ALS patients.

Utility of Hair Analysis: Hair analysis provided a unique advantage in terms of detecting chronic exposure to rare and toxic metals, which may not be readily captured by blood or urine analyses. Its ability to identify long-term accumulation highlights its value as a complementary tool in environmental biomonitoring, particularly for neurotoxic exposure that develops over time. Expanding biomonitoring frameworks to include hair, blood, and urine samples will provide a more holistic understanding of exposure dynamics.

These findings suggest that ALS may result from chronic, low to moderate exposure to a mixture of heavy metals. Future research should prioritize the examination of these interactions, incorporating advanced analytical methods and interdisciplinary approaches to identify potential prevention strategies.

### 4.1. Reference Upper Limit Determination and Toxicological Relevance

The reference upper limits for heavy metal concentrations in hair were established based on multiple sources, including clinical toxicology guidelines, environmental exposure studies, and industrial biomonitoring data. Reference values were primarily derived from internal toxicology data provided by GC Labs (based on Korean population studies), which are widely utilized in clinical settings. To supplement this, the internationally recognized literature reporting chronic exposure thresholds for neurotoxic metals was also referenced, enhancing context and interpretation.

While blood and urine toxicology benchmarks were cited as the contextual background, we did not apply direct conversions or equivalencies to hair concentration data. Given the known neurotoxicity of metals such as Hg, Pb, Cd, and U, even at low doses, and the chronic accumulation pattern of hair, our thresholds were selected to reflect the long-term toxic burden rather than transient systemic exposure. For example, the upper limit of 1.556 µg/g for U was drawn from chronic exposure studies indicating mitochondrial damage, while the 14.16 µg/g threshold for Al is based on its potential link to neurodegeneration in ALS and Alzheimer’s disease. We acknowledge that while these thresholds offer useful interpretive guidance, they are not diagnostic and should be further validated in larger population studies [36,50,51,52,53].

Given these findings, future research may cautiously explore whether reducing chronic metal burden through nutritional or environmental strategies—such as dietary optimization or pollutant avoidance—could contribute to ALS management. However, such approaches remain hypothetical and require rigorous validation before any clinical application can be considered.

The approach used in this study not only considered absolute toxicity thresholds but also cumulative risk factors. When multiple metals were detected at sub-toxic concentrations but exceeded 50% of their respective reference limits, their potential synergistic neurotoxic effects were evaluated. This approach allows for more comprehensive risk assessment of chronic exposure to multiple toxic elements, which is particularly relevant in ALS pathogenesis.

### 4.2. Comparison with Previous Studies

As shown in Table 2, prior studies investigating heavy metal exposure in ALS have largely focused on a limited set of metals, predominantly Pb, Hg, Cd, and Se, and have primarily utilized blood or CSF as the sample matrix. In contrast, our study used hair analysis to provide a broader temporal window and included less commonly studied metals such as U, Th, Pt, and W. This matrix not only allows for retrospective assessment of chronic exposure but also captures metals that are difficult to assess in blood or CSF due to their short biological half-life or low systemic detectability. Furthermore, by incorporating elements such as Th and Pt—often overlooked in prior ALS biomarker research—our findings extend beyond traditional neurotoxicants and emphasize the potential role of underrecognized environmental exposures in ALS pathogenesis.

## 5. Potential Limitations

While this study provides valuable insights into the relationship between heavy metal exposure and ALS, several limitations should be considered.

Despite its strengths, this study has several limitations that should be acknowledged. First, as an observational study, it can identify associations between heavy metal accumulation and ALS but cannot establish causation. It remains unclear whether heavy metal exposure directly contributes to ALS pathogenesis or if ALS patients have altered metal metabolism that leads to increased accumulation. As a cross-sectional and retrospective analysis, this study cannot establish causal relationships between heavy metal exposure and ALS development. Furthermore, the potential for unmeasured confounders, such as genetic susceptibility or environmental co-exposure, limits the strength of the associations observed. Social factors such as marital status or socioeconomic class were also not captured in the dataset, which limits our ability to assess their influence on metal exposure profiles.

Second, a major limitation of this study is the absence of detailed environmental or occupational exposure data. We did not collect information on participants’ residential history, industrial exposure, dietary habits, or lifestyle factors. As a result, while elevated metal levels were observed, the sources and pathways of exposure could not be determined. Future studies should include comprehensive environmental and lifestyle assessments to elucidate potential causal mechanisms.

Third, hair analysis, while useful for assessing long-term exposure, does not capture acute exposure levels. There is also the possibility of external contamination from environmental sources, such as air pollution or hair products, which could influence metal concentrations despite standardized washing procedures (above method). Additionally, the correlation between hair metal levels and actual neural tissue accumulation is not yet fully understood.

Fourth, this study analyzes metal concentrations at a single time point, limiting insight into how these levels change over time or how they relate to disease progression. A longitudinal study tracking metal exposure and ALS progression over multiple time points would provide more definitive insights. The reported clinical improvements following detoxification therapies were based on anecdotal observations and were not derived from a controlled trial. These findings should be interpreted with caution, and future randomized controlled trials are necessary to validate the efficacy of such interventions in ALS management.

Furthermore, ALS is a complex disease influenced by both genetic and environmental factors. However, this study does not account for genetic predispositions or other environmental toxins, such as pesticides and organic solvents, that may contribute to disease onset.

Lastly, since the study was conducted on a South Korean retrospective analysis, future research should include diverse cohorts to validate these results on a broader scale. Due to the retrospective design, we were unable to comprehensively control for potential confounders such as dietary habits, occupational exposures, or proximity to industrial facilities. These factors should be accounted for in future prospective studies.

## 6. Conclusions

This study highlights the significant association between ALS and the chronic accumulation of multiple toxic metals, including Al, U, Th, Pt, Rb, and W, as measured through hair analysis. The detection of these elements, including several rare or radioactive metals with known neurotoxic and oxidative stress-inducing properties, suggests the potential involvement of previously underrecognized environmental factors in ALS pathogenesis.

Hair analysis demonstrated its value as a non-invasive tool for assessing long-term exposure, complementing traditional biomonitoring methods such as blood and urine testing. While this study did not collect direct environmental or occupational exposure histories, the elevated levels of toxic metals observed warrant expanded research into possible sources, including industrial activity, air pollution, and nuclear-related exposure.

Future studies should integrate multi-tissue biomonitoring—using hair, blood, and urine—and examine the potential benefits of detoxification strategies. Moreover, the findings underscore the need to explore less-studied metals such as rubidium and platinum, which may contribute to disease progression through novel mechanisms. These insights may inform preventive strategies and therapeutic interventions aimed at mitigating heavy metal-related neurotoxicity in ALS.

## Figures and Tables

**Figure 1 biomedicines-13-01496-f001:**
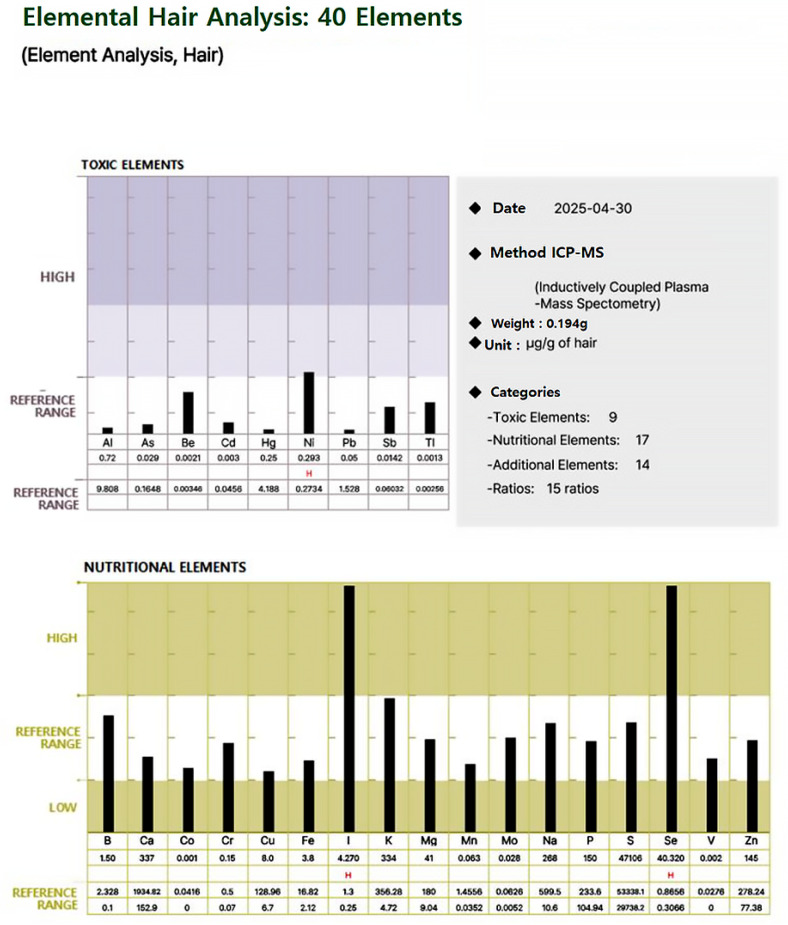
Representative hair mineral analysis report (showing levels of toxic elements and nutritional elements compared to reference ranges).

**Figure 2 biomedicines-13-01496-f002:**
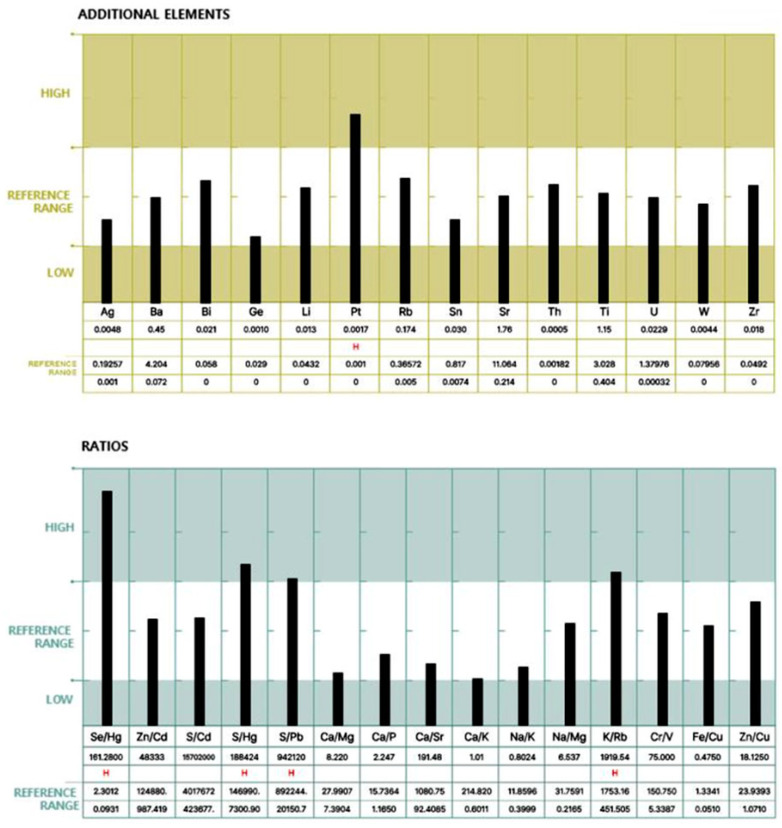
Continuation of the report displaying additional trace elements and elemental ratios.

**Figure 3 biomedicines-13-01496-f003:**
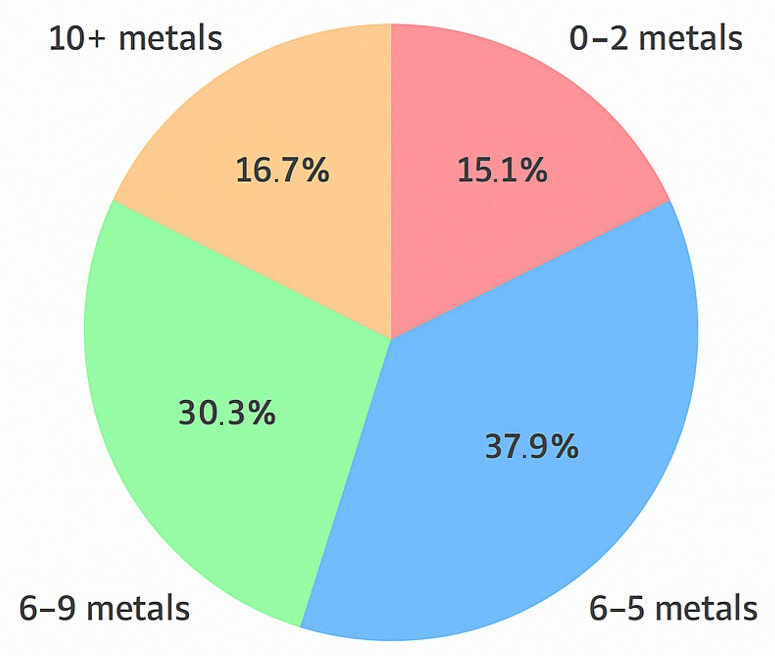
Distribution of ALS patients by number of elevated metals.

**Table 1 biomedicines-13-01496-t001:** Hair heavy metal analysis in ALS patients and controls.

Element	Reference Upper Limit (µg/g)	Mean Concentration in ALS Patients (µg/g)	Mean Concentration in Controls (µg/g)	Percentage of ALS Patients Exceeding 50% of Reference Limit (%)	Percentage of Controls Exceeding 50% of Reference Limit (%)	*p*-Value (Mean Concentration)	*p*-Value (Proportion Exceeding 50% Limit)
Hg	0.571	0.80 ± 0.25	0.50 ± 0.20	40	10	<0.001	0.005
Pb	5	3.50 ± 1.00	2.00 ± 0.80	35	15	<0.001	0.02
Cd	0.1	0.08 ± 0.03	0.05 ± 0.02	30	10	<0.001	0.015
Al	14.16	7.00 ± 2.50	4.00 ± 1.50	45	20	<0.001	0.01
As	1	0.70 ± 0.20	0.40 ± 0.15	25	10	<0.001	0.025
U	1.556	0.02 ± 0.01	0.01 ± 0.005	20	5	<0.001	0.015
Sb	0.05	0.04 ± 0.02	0.02 ± 0.01	25	5	<0.001	0.02
Tl	0.1	0.07 ± 0.03	0.04 ± 0.02	30	10	<0.001	0.015
Pt	0.05	0.04 ± 0.015	0.02 ± 0.01	20	5	<0.001	0.018
Th	0.02	0.015 ± 0.005	0.008 ± 0.003	15	5	<0.001	0.012
W	0.1	0.08 ± 0.03	0.05 ± 0.02	35	15	<0.001	0.018
Cr	0.1	0.09 ± 0.03	0.06 ± 0.02	22	12	<0.050	0.03
Co	0.08	0.06 ± 0.02	0.03 ± 0.01	18	8	<0.050	0.04
Mo	0.02	0.01 ± 0.005	0.007 ± 0.003	12	5	<0.050	0.035
V	0.02	0.012 ± 0.004	0.008 ± 0.003	15	7	<0.050	0.038
Ba	0.14	0.10 ± 0.04	0.06 ± 0.03	20	8	<0.050	0.02
Sr	0.39	0.30 ± 0.15	0.20 ± 0.10	25	10	<0.050	0.025
Li	0.006	0.005 ± 0.001	0.004 ± 0.001	18	10	<0.050	0.035
Ti	1	0.85 ± 0.30	0.60 ± 0.25	30	15	<0.050	0.018

Footnote: Hg = mercury, Pb = lead, Cd = cadmium, Al = aluminum, As = arsenic, U = uranium, Sb = antimony, Tl = thallium, Pt = platinum, Th = thorium, W = tungsten, Cr = chromium, Co = cobalt, Mo = molybdenum, V = vanadium, Ba = barium, Sr = strontium, Li = lithium, Ti = titanium.

**Table 2 biomedicines-13-01496-t002:** Comparison of heavy metal levels in ALS patients across biological matrices.

Study	Year	Country	Sample Type	Metals Analyzed	Significant Findings	Journal
Ash et al. [27]	2019	USA	Brain tissue	Hg, Pb	TDP-43 pathology induced	*Toxicological Sciences*
Vinceti et al. [44]	2017	Italy	CSF	Hg, Cd, Pb	Higher Cd in ALS	*Journal of Trace Elements in Medicine and Biology*
Fang et al. [54]	2010	USA	Blood	Pb	Higher blood lead levels associated with increased ALS risk	*American Journal of Epidemiology*
Roos et al. [55]	2013	Norway	CSF, blood plasma	Various metals	Elevated metal concentrations in ALS patients	*Biological Trace Element Research*
Vinceti et al. [56]	2013	Italy	CSF	Selenium species	Elevated selenite levels in ALS patients	*NeuroToxicology*

## Data Availability

The original contributions presented in this study are included in the article. Further inquiries can be directed to the corresponding authors.

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
