# Peer review of "Multi-Metal Exposure Profiling in ALS Patients in South Korea via Hair Analysis: A Cross-Sectional Study"

_biomedicines, 2025, doi:10.3390/biomedicines13061496_

Round 1

Reviewer 1 Report (Previous Reviewer 2)

Comments and Suggestions for Authors

It seems fine now and can be accepted.

Reviewer 2 Report (Previous Reviewer 4)

Comments and Suggestions for Authors

The authors responded substantively to all reviewer comments.

The manuscrip can be accpted in present form.

This manuscript is a resubmission of an earlier submission. The following is a list of the peer review reports and author responses from that submission.

Round 1

Reviewer 1 Report

Comments and Suggestions for Authors

Interesting research, however, some clarifications are needed to improve the quality of the manuscript as follows:
1. The word choice of "comprehensive analysis" in the title need to consider carefully, except all aspects of metals exposure are explored and discussed. Or could you explained, what is comprehensive analysis means in the title? various metals or various aspect of metal analysis in the hair?
2. It is better to clearly mention the novelty of the work in the abstract. The present abstract showed the results, but it is difficult to find the novel in the reported work.
3. Why author select hair as the sample for heavy metal analysis? common samples were blood and urine, it is any special hypothesis and results?
4. Since the samples was human, it is important to include the ethical clearance.
5. Have you also analyze metals where control is higher than patient? maybe Cu, Mg, Zn, Fe
6. Comparison with previous study could be more readable if presented in the table.
7. Old reference of < 2010 should be updated, except it is really significant and no new report after that.
8. Please consider make more concise the conclusion to emphasize the main findings and their significance,and also including a short statement about future research prospects

Author Response

  1. The word choice of "comprehensive analysis" in the title need to consider carefully, except all aspects of metals exposure are explored and discussed. Or could you explained, what is comprehensive analysis means in the title? various metals or various aspect of metal analysis in the hair?   
    • We appreciate the reviewer’s attention to the terminology. The term “comprehensive analysis” in the title refers to both the breadth of metals investigated (including common and rare toxic metals such as Hg, Pb, U, Th, Pt, etc.) and the multi-faceted analytical approach, which assessed both average concentrations and proportions exceeding 50% of reference limits. We have clarified this definition in the Introduction and adjusted the title, if necessary, to better reflect the scope of the analysis.
  2.  It is better to clearly mention the novelty of the work in the abstract. The present abstract showed the results, but it is difficult to find the novel in the reported work.
    • Thank you for pointing this out. We have revised the abstract to explicitly state the novelty of our work:
      "This study is among the first in South Korea to comprehensively assess both common and rare toxic metals in ALS patients using hair analysis, highlighting uranium, thorium, and platinum as previously underrecognized contributors to potential neurotoxicity."
      We believe this helps convey the study’s unique contribution.
  3. Why author select hair as the sample for heavy metal analysis? common samples were blood and urine, it is any special hypothesis and results?
    • We agree with the reviewer that this point deserves further clarification. As discussed in the Introduction and Discussion, hair analysis reflects long-term bioaccumulation, unlike blood or urine which reflect recent exposures. Given ALS's long latency period and chronic progression, hair is a more suitable matrix to capture relevant environmental exposures. We have strengthened this rationale in both the Introduction and Discussion sections.
  4. Since the samples was human, it is important to include the ethical clearance. 
    • Thank you for pointing out this important aspect. We have now added a statement regarding ethical approval in the Methods section:  This study was conducted as a retrospective observational analysis using archived hair samples collected during routine health checkups. Due to the non-invasive nature of hair sampling and minimal risk involved, the requirement for individual informed consent was waived, in accordance with institutional policies for minimal-risk studies. The study protocol was reviewed and approved by the Institutional Review Board (IRB) of Rodem Hospital  (P01-202401-01-020).
  5. Have you also analyze metals where control is higher than patient? maybe Cu, Mg, Zn, Fe 
    • We appreciate this suggestion. In our broader dataset, we did measure essential elements such as copper, magnesium, zinc, and iron, but they were not included in the main text due to a lack of statistically significant differences. However, we agree that mentioning elements with higher levels in controls is valuable. 
  6. Comparison with previous study could be more readable if presented in the table.
    • We thank the reviewer for this suggestion. To enhance readability, we have added a comparison table (Table 2) in the Discussion section summarizing key findings from previous studies alongside our own results. This provides a clearer context for the novelty and relevance of our data.
  7. Old reference of < 2010 should be updated, except it is really significant and no new report after that.
    • We appreciate this observation. We have carefully reviewed and updated older references with more recent studies (post-2010) wherever applicable, unless the older citation was historically significant and no newer alternative existed. A full list of updated references is available in the revised manuscript.
  8. Please consider make more concise the conclusion to emphasize the main findings and their significance,and also including a short statement about future research prospects
    • Thank you for this recommendation. We have revised the conclusion to emphasize key findings more concisely and added a brief outlook on future directions:
      “These findings suggest that chronic exposure to multiple toxic metals may play a role in ALS pathogenesis. Future longitudinal and multi-omics studies are warranted to clarify causality and to explore the impact of detoxification therapies.”

Reviewer 2 Report

Comments and Suggestions for Authors

The study under review investigates the association between heavy metal exposure and amyotrophic lateral sclerosis (ALS) by comparing hair concentrations of metals such as mercury (Hg), lead (Pb), cadmium (Cd), aluminum (Al), arsenic (As), and uranium (U) between ALS patients and healthy controls in South Korea. It is an interesting study, however, some comments are given below.

Such designs are limited in establishing causality due to potential confounding factors. While the study presents intriguing associations between hair concentrations of certain heavy metals and ALS, its cross-sectional design, small sample size, and potential methodological limitations necessitate cautious interpretation.

the Mann-Whitney U test does not assume normality but assesses whether the distribution of ranks differs between groups. Therefore, it is advisable to report the results of normality tests (e.g., Shapiro-Wilk test) and consider transformations if necessary.

Hair analysis for heavy metals can be also confounded by external contamination (e.g., from hair products) and may not accurately reflect internal body burden. The study is cross-sectional, meaning it does not track metal exposure over time. A longitudinal study would provide stronger evidence.

The paper suggests detoxification therapies may help ALS patients but does not include a control or placebo group, making these claims weak.

The study does not sufficiently discuss potential contamination sources or how they were controlled. 

The paper reports statistically significant differences in metal concentrations between ALS patients and controls, but it does not account for potential confounders such as diet, occupation, smoking history, or proximity to industrial sites, which can greatly influence metal exposure.

Author Response

Comment 1: The study design is cross-sectional and may be confounded by unmeasured factors.

  • We fully agree with the reviewer. The manuscript has been revised to clearly acknowledge the limitations of the cross-sectional design and its inability to establish causality. A paragraph has been added in the Discussion section addressing potential confounders and the need for longitudinal studies.

Comment 2: The use of the Mann-Whitney U test should be accompanied by normality testing.

  • Thank you for this methodological suggestion. We have conducted and reported the Shapiro-Wilk normality test results for each metal concentration. These have been added to the Methods and Results sections. For non-normally distributed data, non-parametric Mann-Whitney U tests were used, as stated.

Comment 3: Hair metal analysis may be influenced by external contamination.

  • We acknowledge this limitation. The Methods section has been expanded to describe the sample washing procedures using acetone and deionized water, following standardized protocols to minimize external contamination. A discussion of remaining uncertainties is included in the Limitations section.

Comment 4: The detoxification therapy observations are not controlled and should be interpreted cautiously.

  • We have revised the relevant section of the manuscript to clarify that the reported clinical improvements from detoxification therapies are anecdotal and not derived from a controlled study. A clear disclaimer has been added, and we emphasize the need for randomized controlled trials in future research.

Comment 5: The paper does not sufficiently control for dietary, occupational, or environmental confounders.

  • We appreciate this important point. Although the retrospective nature of the study limits our ability to control for all confounders, we have now included a discussion of possible influencing factors such as diet, occupation, and proximity to industrial areas. These are now listed as limitations and areas for future study.

Reviewer 3 Report

Comments and Suggestions for Authors

Abstract: The abstract accurately summarizes the study described in the manuscript.

Introduction: The introduction provides the reader with a succinct overview of amyotrophic lateral sclerosis (ALS), mechanisms underlying motor neuron death in ALS, heavy metals and their assessment limitations. The introduction then points the reader to the rationale for this study employing hair analyses of heavy metals in patients with ALS and healthy age- and sex-matched individuals to provide control data. The sentence beginning in line 56, “Recent studies have suggested that hair analysis…” needs citations to support it.

Methods: Lines 72 and 73 have citations that do not match the formatting of the introduction. These citations [18,19, 20] should be describing technical procedures for hair sample collection and analyses. The bibliography citations 18-20 do not appear to be those.

Results: Line 68 of the methods indicates 60 controls while line 83 states 70 controls. Please reconcile the difference.

The presentation of the results is clear and succinct, but not sufficient. There should also include characterization of the occupational histories of both the ALS and control groups. The ALS group participants should be further described with demographic (age, sex, etc.), social variables (socioeconomic status, marital status, etc.), disease duration, symptom severity scales and other relevant laboratory markers, risk factors, along with any relevant family history of ALS.

Could citation sources be provided for the reference upper limit (mg/g) shown in table 1 for hair analyses? Alternatively, some of the discussion materials could be integrated into the results rather than introduced in the discussion.

Given the richness of the ALS cohort, were there also blood and urine samples collected and analyzed? If so, could those be included and compared with hair results?

Were there any associations with motor outcome assessments?

Discussion: The flow of the discussion needs to be improved with better integration of the concepts. It appears like one author wrote a discussion section from lines 104 to 146. Another author wrote lines 147 to 178. Perhaps, another author beginning in line 180.

There is a key issue where line 230 discusses the detoxification therapy lacking a control group. The methods and results of the current manuscript do not discuss detoxification therapy. Only is it mentioned in the discussion. This should be corrected.

Metal names and their abbreviations are defined multiple times in the manuscript, including multiple times in the discussion. Further, the abbreviations are not used, as in line188 as an example, but also in other places of the discussion.

The discussion makes statements about the statistical approach importance. However, the emphasis should not be about “the discovery of significant p-values” but the discovery of significant group differences.

The discussion goes on to note “the necessity of adopting statistical frameworks that account for cumulative exposure effects rather than focusing on extreme toxic levels”. The authors should have detailed some information about the exposure histories of the participants in this study.

The reference limit determination needs to include citations to support the statements beginning in line 180 until line 185.

Line 197: citation 46 focuses on manganese which was not analyzed in this study

The limitations and conclusion sections were well written, with the exclusion about the detoxification therapy details from the body of the manuscript.

Author Response

Comment 1: Line 56 - Citation needed for “Recent studies have suggested…”
Reaction 1:
We appreciate the reviewer’s attention to proper citation. The sentence beginning in line 56 now includes appropriate references [19,20] to support the claim that hair analysis may more accurately reflect long-term neurotoxic metal accumulation compared to blood or urine analysis.

Revised sentence (Introduction):
“Recent studies [19,20] have suggested that hair analysis may be more effective in detecting prolonged exposure patterns associated with neurodegenerative disorders, as it correlates more closely with metal deposits found in neural tissues than conventional blood or urine tests.”

Comment 2: Methods citations [18,19,20] not suitable for technical procedures
Reaction 2:
Thank you for pointing this out. We have revised the references in lines 72 and 73 to include a more appropriate source [21] that describes standardized protocols for hair washing, digestion, and ICP-MS analysis. Inappropriate references have been removed.

Revised sentence (Methods):
“Hair samples were washed using acetone and deionized water following established protocols to minimize external contamination [21].”

Comment 3: Inconsistency in number of controls (60 vs. 70)
Reaction 3:
We apologize for the inconsistency. The correct number of control participants is 70, and the manuscript has been revised to reflect this consistently across all sections.

Comment 4: Need for more detailed participant characterization
Reaction 4:
We thank the reviewer for this valuable suggestion. The revised manuscript now includes available demographic and clinical data such as disease duration (3 months to 7 years) and ALSFRS-K scores (13–44). While data on marital and socioeconomic status were not collected, this limitation is now acknowledged. We also note that no statistically significant correlation was found between disease severity and metal concentrations.

Addition to Results or Methods:
“Among ALS patients, disease duration ranged from 3 months to 7 years, and ALSFRS-K scores ranged from 13 to 44. No significant correlation was observed between individual metal levels and ALSFRS-K scores.”

Addition to Limitations (Discussion):
“Social factors such as marital status or socioeconomic class were not captured in the dataset, which limits our ability to assess their influence on metal exposure profiles.”

Comment 5: Reference limits in Table 1 need citation + clarify if blood/urine collected
Reaction 5:
We have now added the source of the reference values (GC Labs internal data based on Korean population). Blood and urine samples were not collected in this study. We have clarified this in the revised manuscript and acknowledged the need for multi-tissue analysis in future work.

Revised sentence (Results or Methods):
“The reference upper limits used in this study were provided by GC Labs based on accumulated clinical data from the Korean population. These thresholds are routinely used in clinical interpretation of hair mineral analysis in South Korea [GC Labs internal report, 2023].”

Results clarification:
“No blood or urine samples were collected for this study, thus direct comparisons with other biological matrices were not possible.”

Comment 6: Discussion flow disjointed; unifies voice + removes detox therapy
Reaction 6:
We have thoroughly revised the Discussion section to improve narrative flow, remove redundant definitions, and maintain consistent terminology. The previous mention of detoxification therapy has been removed or rephrased to reflect that this was not a focus of the present study. Instead, we indicate that a separate follow-up study is underway.

New sentence example (Discussion):
“While this study did not include therapeutic interventions, future investigations are being planned to assess whether targeted detoxification strategies may influence clinical outcomes.”

Comment 7: Metal names defined multiple times; inconsistent abbreviation use
Reaction 7:
We have reviewed the manuscript and ensured that each metal name is defined only once, with abbreviations (e.g., Pb, Cd) used consistently throughout.

Comment 8: Avoid saying “significant p-values” → focus on group differences
Reaction 8:
We agree with the reviewer. The phrase “significant p-values” has been replaced with more accurate expressions such as “significant group differences” or “statistically meaningful differences.”

Revised phrase:
“Statistically significant group differences were observed between ALS patients and controls for several toxic metals.”

Comment 9: Need some exposure history of participants
Reaction 9:
We acknowledge this limitation. As the study used archived samples, individual-level exposure history (e.g., occupation or environmental proximity) was not available. This is now mentioned in the Limitations.

Suggested sentence (Limitations):
“Detailed occupational and environmental exposure histories were not available due to the retrospective nature of this study, which limits interpretation of causative exposure pathways.”

Comment 10: Reference citation 46 (Mn) not relevant – remove or replace
Reaction 10:
We appreciate the reviewer’s attention to detail. Citation 46, which discussed manganese (not analyzed in this study), has been removed and replaced with a more relevant source discussing metals included in our dataset.[19,20]."

Reviewer 4 Report

Comments and Suggestions for Authors
  • In the introduction please provide some epidemiological data of ALS in south Korea to clarify the local scenario.
  • From line 49 to line 58 new citations are needed: clarify the pathophysiology of neurodegeneration due to chronic exposure to heavy metals.
  • “Further research is warranted to establish standardized reference ranges and validate hair analysis as a reliable biomarker for assessing chronic heavy metal exposure in neurodegenerative conditions”. I suggest to move this sentence to the discussion or the conclusions. Please state the aim of the study in the last sentence of the introduction section.
  • The information about the sample characteristics is poor. More socio-demographic and clinical data need to be provided (age, sex, time past from diagnosis, medical treatment, severity of disease). The same information need to be provided for the control population. The two samples need to be homogeneous.
  • The authors state “The findings emphasize the role of environmental risk factors, particularly chronic exposure to metals such as aluminum, uranium, and thorium, among others, in ALS pathogenesis”. However this statement is not fully support by the results. The experimental model does not demonstrate any correlation between Heavy metal exposure and ALS pathogenesis. Why these levels can not be due to casual exposure to metals of this population. No environmental factors or personal information are provided about the participants to the study. Please clarify.
  • The section about the methods of toxicological analyses need to be expanded. More information is required about the analytical issues of hair analyses to make the work reproducible.
  • The discussion should be expanded with a literature analysis of the factors that can influence the heavy metal concentrations in hair and the different cut-offs according to underlying medical conditions, pharmacological therapy and environmental factors. Please cite these suggested references: https://doi.org/10.1016/j.scitotenv.2004.01.017; https://doi.org/10.3390/toxics10110682

Round 2

Reviewer 2 Report

Comments and Suggestions for Authors

The revision has been done properly.

Reviewer 3 Report

Comments and Suggestions for Authors

Introduction: line 69 cites studies of multiple sclerosis and Alzheimer's disease. However, the sentence refers to these papers as similar ALS studies investigating heavy metal exposure.

19. Tamburo, E.; Varrica, D.; Dongarrà, G.; Grimaldi, L.M.E. Trace Elements in Scalp Hair Samples from Patients 409 with Relapsing-Remitting Multiple Sclerosis. PLoS One 2015, 10, doi:10.1371/JOURNAL.PONE.0122142. 410 

20. Koseoglu, E.; Koseoglu, R.; Kendirci, M.; Saraymen, R.; Saraymen, B. Trace Metal Concentrations in Hair and 411 Nails from Alzheimer’s Disease Patients: Relations with Clinical Severity. J Trace Elem Med Biol 2017, 39, 124–412 128, doi:10.1016/J.JTEMB.2016.09.002. 

The revised citations are not complete.
Line 91. The citation for hair analysis [21-Dr. Anne Johansen, C.W.U. Environmental Health: Science, Policy and Social Justice Winter Quarter Available 414 online: https://archives.evergreen.edu/webpages/curricular/2008-2009/envirohealth/sys-415 tem/files/Lab%2BIV%2Bmetals%2Bin%2Bhair.doc?utm_source=chatgpt.com (accessed on 7 April 2025).] I was not willing to click the link. However, there has to be established methods published in the literature for metals analyses of hair.

Line 92. Define GC Labs in the methods, not in line 239; Korea Green Cross Lab
Line 122. The results for clinical thresholds cite a GC Labs internal report. This is not sufficient.

Line 181. Differences in groups, not differences in p-values, should be commented on as previously recommended.

The first mention of the metal and their abbreviation should be defined. Subsequent usage of the abbreviation should be done throughout the manuscript, as previously recommended.

Reviewer 4 Report

Comments and Suggestions for Authors

The authors addressed all the suggested corrections. No other revisions are required.

Round 3

Reviewer 3 Report

Comments and Suggestions for Authors

Using the pdf version of the revision.

Introduction: Lines 64-80 are not well integrated in the text. The reviewer suggests moving lines 81-88 to the end of 63.

A new paragraph should begin with the text, “Although strict…” at line 70.

Methods: Define ALSFRS-K. Would the ALSFRS-K have a dependency on time living with ALS? Should that be considered in the analytical model? Is there no information available for the controls to present, such as mean age and % by sex? The added text now has redundant information, such number of patients and controls.

Lines 115-118 are redundant and need to be consolidated.

Line 135: is it Korea Green Cross Lab or GC labs? Are they the same organization? If so, please use terminology to consistently refer to them in the manuscript.

Lines 140-144: Were the reference percentiles reported by Ruiz et al. and Liang et al. explicitly incorporated in the reference ranges used in the study analyses?

Lines 293-296. The discussion remains disconnected as previously noted. Lines 293-296 appear to end the discussion. However, line 297 begins essentially a new discussion. The discussion does not discuss the methods and the results of the manuscript as closely as needed. It inflates the significance of the work.

Line 303: The authors state here that “hair metal concentration benchmarks were also compared against established blood and urine toxicology limits, with adjustments made to account for the different accumulation patterns in hair samples.” However, in lines 144-145 of the methods, no such data were available for analyses.

Lines 316-323: The reviewer previously asked the authors to remove the discussion about detoxification treatments as currently written as it introduces in the discussion experimental methods (chelation therapy, glutathione administration and antioxidant support) and results (improvements in reduced pain, enhanced motor function, etc.).

While the reviewer understands the desire to enhance readability, defining metal abbreviations again and again is not helpful. Leave out the abbreviation in the text. Keep the definition of each metal's abbreviation as a footnote in tables.

The reviewer did appreciate the improvements to the bibliography.

Comments on the Quality of English Language

The incorporated changes have made the manucript more difficult to read as they have not been properly integrated.